# Strengthening health systems to improve the value of tuberculosis diagnostics in South Africa: A cost and cost-effectiveness analysis

Nicola Foster [1,2,3]*, Lucy Cunnama [1], Kerrigan McCarthy [4,5], Lebogang Ramma[6], Mariana Siapka[3], Edina Sinanovic[1], Gavin Churchyard[5,7], Katherine Fielding[3,5], Alison D. Grant[3,5,8], Susan Cleary[1]

1 Health Economics Unit, School of Public Health and Family Medicine, University of Cape Town, Cape Town, South Africa, 2 Division of Health Research, Lancaster University, Lancaster, United Kingdom, 3 TB Centre, London School of Hygiene & Tropical Medicine, London, United Kingdom, 4 Division of Public Health, Surveillance and Response, National Institute for Communicable Disease of the National Health Laboratory Service, Johannesburg, South Africa, 5 School of Public Health, Faculty of Health Sciences, University of the Witwatersrand, Johannesburg, South Africa, 6 Department of Health and Rehabilitation Sciences, University of Cape Town, Cape Town, South Africa, 7 Aurum Institute, Johannesburg, South Africa, 8 Africa Health Research Institute, School of Nursing and Public Health, University of KwaZulu-Natal, Durban, South Africa

* nicola.foster@lshtm.ac.uk

**Data Availability Statement:** All relevant data are within the paper and it's Supporting information files.

## Abstract

### Background

In South Africa, replacing smear microscopy with Xpert-MTB/RIF (Xpert) for tuberculosis diagnosis did not reduce mortality and was cost-neutral. The unchanged mortality has been attributed to suboptimal Xpert implementation. We developed a mathematical model to explore how complementary investments may improve cost-effectiveness of the tuberculosis diagnostic algorithm.

### Methods

Complementary investments in the tuberculosis diagnostic pathway were compared to the status quo. Investment scenarios following an initial Xpert test included actions to reduce pre-treatment loss-to-follow-up; supporting same-day clinical diagnosis of tuberculosis after a negative result; and improving access to further tuberculosis diagnostic tests following a negative result. We estimated costs, deaths and disability-adjusted-life-years (DALYs) averted from provider and societal perspectives. Sensitivity analyses explored the mediating influence of behavioural, disease- and organisational characteristics on investment effectiveness.

### Findings

Among a cohort of symptomatic patients tested for tuberculosis, with an estimated active tuberculosis prevalence of 13%, reducing pre-treatment loss-to-follow-up from ~20% to ~0% led to a 4% (uncertainty interval [UI] 3; 4%) reduction in mortality compared to the Xpert scenario. Improving access to further tuberculosis diagnostic tests from ~4% to 90%

**Funding:** The XTEND project was carried out with the support of the Bill and Melinda Gates Foundation (Grant OPP1034523). This analysis contributes to NF's PhD, funded by the Medical Research Council of South Africa in terms of the National Health Scholars Programme from funds provided by the Public Health Enhancement Fund. This paper was prepared with support from the Collaboration for Health Systems Analysis and Innovation (www.chesai.org) that receives funding from the International Development Research Centre Ottawa Canada. The funders had no involvement in the study design; in the collection, analysis and interpretation of data; in the writing of the report; and in the decision to submit the paper for publication.

**Competing interests:** The authors have declared that no competing interests exist.

among those with an initial negative Xpert result reduced overall mortality by 28% (UI 27; 28) at $39.70/ DALY averted. Effectiveness of investment scenarios to improve access to further diagnostic tests was dependent on a high return rate for follow-up visits.

## Interpretation

Investing in direct and indirect costs to support the TB diagnostic pathway is potentially highly cost-effective.

## Introduction

Globally, there is renewed interest in understanding how disease-specific investments function in the context of broader health system challenges [1]. Alongside this interest is re-invigorated enquiry into how best to support policy makers to assess joint technology and health systems strengthening investments when introducing new technologies. A recent example of an investment with global importance is the roll-out of Xpert MTB/RIF (Xpert).

In 2011, South Africa started the national roll-out of Xpert as first-line tuberculosis diagnostic test, following the World Health Organization (WHO) recommendation [2]. The roll-out was anticipated to result in more people starting tuberculosis treatment because of Xpert's higher sensitivity, thus reducing mortality [3]. In addition Xpert was expected to reduce the time to MDR tuberculosis treatment start [4, 5]. However, in practice no significant impact on tuberculosis-related morbidity, mortality, pre-treatment loss-to-follow-up (iLTFU) or time-to-treatment for patients starting drug-sensitive tuberculosis (DS-TB) has been observed [6, 7]. Studies examining the impact on patients with multi-drug resistant (MDR) tuberculosis found that Xpert reduced time-to-appropriate-treatment, although not to same day or same week, as had been expected [8, 9]. Furthermore, an economic evaluation based on a pragmatic trial following the roll-out in South Africa (the XTEND trial) found that Xpert implementation was both effect- and cost-neutral and was unlikely to improve the cost-effectiveness of the tuberculosis diagnostic algorithm [10]. The study concluded that implementation constraints may have mediated the impact of Xpert under programmatic conditions [7]. Other countries reported similar experiences with Xpert implementation. Placement of the test in the health system, it's integration into the laboratory infrastructure and diagnostic algorithm, as well as patient linkages to treatment were found to be important mediators of costs and effects [11–18].

For South Africa and beyond, policy makers need support to determine which complementary investments are required to strengthen the tuberculosis diagnostic pathway. To inform this need and illustrate a potential approach to assessing combined diagnostic technology and health systems investments, we fitted a purpose-built mathematical model to empirical data from the XTEND trial [7]. We then explored which investments complementary to the Xpert-based tuberculosis diagnostic algorithm would be most cost-effective in South Africa, and used the model to identify drivers of the cost-effectiveness of these investments.

## Methods

We conducted cost-effectiveness analyses of investments in health systems to support tuberculosis diagnosis. This analysis builds on previous modelling work that explored investments in

patient pathways [19–22], by using patient-level cohort data from a pragmatic cluster-randomised controlled trial (described in S1 Text).

## Overview

Health systems investments are typically conceptualised as investments in health care infrastructure, clinical guidelines, technology or human resources, with less emphasis on how the relational aspect of health systems [23] may affect the costs and outcomes of an investment. Clinical discretionary decision-points in patient care can be conceptualised as transactions between patients and providers, occurring within a given organisational system. One may consider these transactions as interactions between the hardware (technology, infrastructure and finances) and software (formal or informal rules of practice, and beliefs that explain behaviour) components of health systems [24]. While investment costs have been estimated by analysing how the production of healthcare responds to an increase in need [25, 26], here we identified patterns of provider behaviour and then modelled this behaviour as a function of resource availability, process and relational interactions. This is implemented in the model by the mediation of decisions along the patient pathway. A simplified visual representation of the model and the decision points is shown in Fig 1 and is referred to in Table 2 [27]. The costs of decision-making processes includes the cost of regulating the decision as well as the opportunity cost of the benefits forgone in the time taken to make the decision or in making the wrong decision, the transaction cost [28: 86]. We modelled the value of additional investments to strengthen these decision-making processes.

**Mathematical model.**   A state-transition model with time-dependent Markov processes was developed, simulating disease progression and interactions with the health system in a symptomatic population being investigated for tuberculosis. Secondary benefits to the

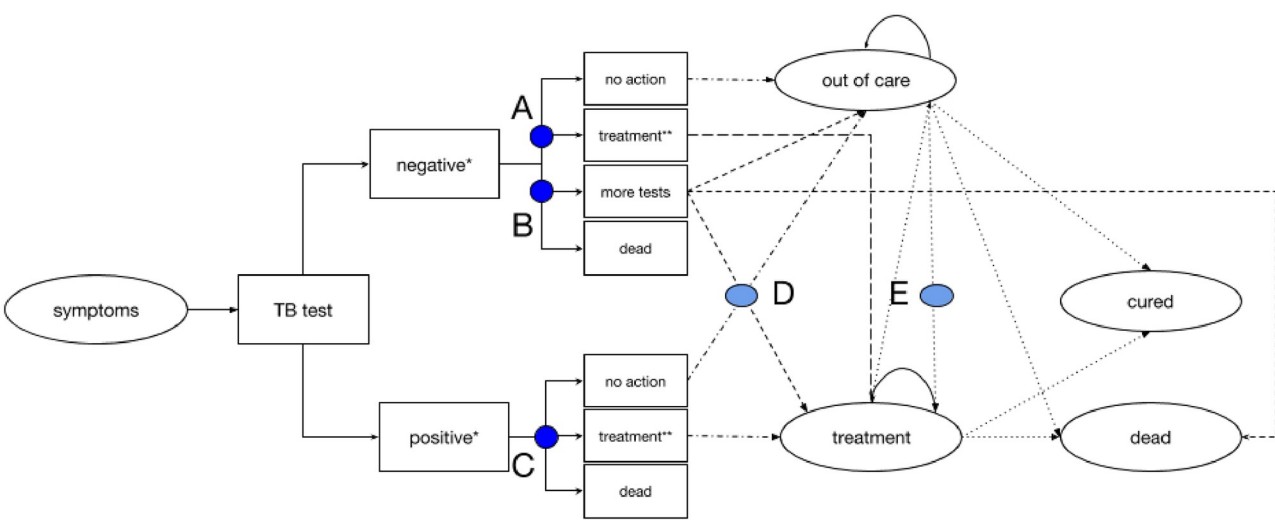

**Fig 1. Simplified schematic of the model.** The figure is a simplified representation of the model, with the circular boxes representing health states and the square boxes representing intermittent states used to model shorter time step process. The model structure is presented in more detail in S1 Text. "A" refers to the decision-making process from a negative test result to starting treatment without bacteriological confirmation; "B" represents the decision to continue testing for tuberculosis (negative pathway) in those with a negative test result; "C" is the behaviour around starting treatment after a positive test results; "D" refers to the decision (based on an interpretation of the further diagnostic tests) to start treatment; and "E" refers to the decision to start tuberculosis (TB) treatment after being 'out of care'. * The model structures following the tuberculosis test are replicated for each of the six patient types, those HIV negative (with and without tuberculosis), HIV positive not on antiretroviral therapy (with and without tuberculosis), and those HIV positive on antiretroviral therapy (with and without tuberculosis). ** The treatments states are replicated for drug-sensitive and multi-drug resistant tuberculosis treatment.

population due to tuberculosis transmission reduction are not included [29]. The analytical timeframe is three years, representing the time until the population is either cured of this tuberculosis episode, or dead. Patients move through the model in one-monthly steps to represent movement through treatment and from out of care, with additional structure added to model shorter diagnostic processes. The model was implemented in TreeAge Pro 2018 and datasets analysed using STATA 13.

In the model, six patient types defined by HIV, anti-retroviral therapy and true tuberculosis status move through health states until reaching an absorbing state (cure or death). Patients are symptomatic when entering the model, transition through a series of diagnostic processes, and then move to one of four possible health states (1) 'out of care' if not started on treatment; (2) drug-sensitive or multi-drug resistant tuberculosis treatment; (3) death; or (4) cured. The 'cured' state can be entered either after treatment or based on a self-cure rate.

### Parameter estimation and model fitting

Transition probabilities and resource use were estimated from trial data (see Table 1). Where treatment-related events occurred after the six-month trial period, data from published cohorts and meta-analyses were used to construct the patient pathway until the end of the treatment episode. The pragmatic nature of the trial did not allow for definitive confirmation of TB diagnosis among trial participants. Unobservable parameters include the true TB prevalence in the population, and the predictive value of decisions to start treatment or request further investigations. These parameters were estimated by calibrating the model's mortality and treatment outputs against those observed in the trial [45]. We estimated a plausible range of values for the unobserved parameters and then iteratively fitted the mortality and time-to-treatment curves from model outputs to trial outcomes until the shape of the respective curves fitted using a range of goodness-of-fit measures [46: 260] (see S1 Text).

### Cost analyses

The costs of providing and accessing care were estimated alongside the trial, using a combination of top-down and ingredients costing approaches [10, 47, 48]. HIV-care costs were extracted from published sources (Table 1). Costs were estimated by multiplying unit costs by the number of events incurred from data collected during the trial. Patient costs included travel- and time-costs incurred by patients and caregivers when accessing care. Additionally, income loss, the cost of caregiver's time, interest on loans as well as the cost of nutritional supplements were included. The opportunity cost of time was valued by multiplying time loss by the pre-illness mean income of the cohort [44]. All costs were estimated in local currency using 2013 prices and converted to US dollars using the average 2013 exchange rate of US $1 = R9.62 (www.Oanda.com).

### Investments

The pragmatic nature of the trial allowed us to identify gaps between ideal movement along different decision nodes of the pathway and mediating variables of effectiveness in routine care settings. Table 2 summarises the investment scenarios modelled and how they were implemented in the model, with a visual representation of the model and decision points provided in Fig 1. We modelled five investment strategies to support the tuberculosis diagnostic pathway. These included 1) reducing initial pre-treatment loss-to-follow-up (iLTFU), 2) supporting same-day clinical diagnosis of tuberculosis after a negative test result (TfN), and 3) improving access to further tuberculosis diagnostic tests following an initial negative result (NP). In addition, two combination scenarios were modelled (iLTFU and TFN; iLTFU and

**Table 1. Summary of parameters and distributions.**

| Definition | Mean and stratification | Distribution | Comments. References are listed as name of first author, year (Reference). |
|---|---|---|---|
| *Population* | | | |
| Gender | 59.9% female | | Represents trial population. Churchyard. 2015 [7] |
| Age (IQR) | 37 (29–48) years | Fixed | Represents trial population. Churchyard. 2015 [7] |
| Initial population disease characteristics | HIVneg 0.314 (0.030); | Dirichlet | From trial population. Churchyard 2015 [7] Those with unknown self-reported HIV status are assumed to be HIV positive, not on ART. |
| | HIVpos 0.531 (0.015); | | |
| | ART 0.155 (0.005) | | |
| CD4 count in those with HIV (IQR) | 315 (192–480) cells/μL | | Represents the microscopy arm of the trial population. Churchyard. 2015 [7] |
| True TB prevalence (includes bacteriologically confirmed -, clinical—and undiagnosed TB) in the microscopy arm of the study. | 13.0% | Fixed | Estimated from XTEND trial and model calibration. Churchyard. 2015 [7] |
| Proportion of patients diagnosed with drug-resistant TB, any diagnosis | 4.0% (8/195) | | Represents trial population. Churchyard. 2015 [7]. |
| Proportion of patients starting MDR-TB treatment | 2.0% (3/195) | | Represents what was observed in the XTEND trial. Churchyard. 2015 [7]. Time to starting MDR TB treatment was 11 and 33 days respectively. |
| *Diagnosis, transition probabilities* | | | |
| Probability of a positive Xpert test result if symptomatic and able to provide a sputum sample, mean (standard deviation) | HIVneg 0.077 (0.03); | Dirichlet | Estimated from XTEND trial. Churchyard. 2015 [7] |
| | HIVpos 0.132 (0.05); | | |
| | ART 0.135 (0.03) | | |
| Probability of TB if patient had a positive test result | HIVneg 0.877; | Fixed | Estimated based on GX sensitivity 0.86 in HIVneg; 0.79 in HIVpos, 0.94 for Rif resistance, and GX specificity of 0.99 in HIVneg, HIVpos, 0.98 for Rif resistance. Steingart 2014 [30], Steingart 2006 [31], and Boehme 2011 [32]. |
| | HIVpos 0.936; | | |
| | ART 0.938 | | |
| Probability of TB if patient had a negative test result | HIVneg 0.012; | Fixed | Unobserved parameter, estimated from model calibration. Based on GX sensitivity 0.86 in HIVneg; 0.79 in HIVpos, 0.94 for Rif resistance, and GX specificity of 0.99 in HIVneg, HIVpos, 0.98 for Rif resistance. Steingart 2014 [30], Steingart 2006 [31], and Boehme 2011 [32]. This includes a probability of a false negative test result; HIVneg 0.012; HIVpos pre-ART 0.038; HIVpos ART 0.039 as well as a probability of 'undiagnosed TB'. Undiagnosed TB includes those who provide pauci-bacillary sputum or have extra-pulmonary TB. Probability of undiagnosed, "hard-to-diagnose" TB estimated to be 0.075 in those HIVpos pre-ART and 0.075 those HIVpos. |
| | HIVpos 0.113; | | |
| | ART 0.114 | | |
| Probability of starting treatment within 30 days of a positive test result, mean (standard deviation) | HIVneg 0.882 (0.325); | Dirichlet | Estimated from XTEND trial. Churchyard et al. 2015 [7] |
| | HIVpos 0.802 (0.400); | | |
| | ART 0.944 (0.236) | | |
| Probability of starting treatment within one month of a negative test result without further diagnostic tests | HIVneg_TB 0.535; HIVneg 0.002; HIVpos_TB 0.072; HIVpos 0.009; ART_TB 0.017; ART 0.003 | Fixed | Probability of starting treatment was estimated from XTEND trial, whether this clinical decision was correct (treatment started in those with TB vs those without) was estimated through model calibration. Churchyard et al. 2015 [7]. We therefore assume that clinicians are unlikely to start treatment empirically in those HIV negative. |
| Probability of receiving further investigations after a negative test result | HIVpos_TB 0.041; HIVpos 0.041; ART_TB 0.073; ART 0.073 | Fixed | Estimated from XTEND trial. Churchyard. 2015 [7]. McCarthy. 2016 [33]. |
| Probability of starting TB treatment after further diagnostic tests | HIVpos_TB 0.212; HIVpos 0.027; ART_TB 0.217; ART 0.037 | Fixed | Estimated from XTEND trial and the model calibration. Churchyard. 2015 [7]. McCarthy. 2016 [33]. |
| Probability of starting TB treatment from 'out of care', by month: from all who do not start TB treatment within one month of the diagnostic test | | | |

*(Continued)*

**Table 1.** (Continued)

| Definition | Mean and stratification | Distribution | Comments. References are listed as name of first author, year (Reference). |
|---|---|---|---|
| Month 2 | HIVneg_TB 0.928; HIVneg 0.005; HIVpos_TB 0.164; HIVpos 0.000; ART_TB 0.100; ART 0.000 | Fixed | Curve estimated from XTEND trial. Assume that the behaviour from out of care remains the same. Churchyard. 2015 [7]. |
| Month 3 | HIVneg_TB 0.756; HIVneg 0.000; HIVpos_TB 0.066; HIVpos 0.000; ART_TB 0.207; ART 0.000 | Fixed | Curve estimated from XTEND. Assume that the behaviour from out of care remains the same. Churchyard. 2015 [7]. |
| Month 4 | HIVneg_TB 0.000; HIVneg 0.005; HIVpos_TB 0.146; HIVpos 0.000; ART_TB 0.148; ART 0.000 | Fixed | Curve estimated from XTEND. Assume that the behaviour from out of care remains the same. Churchyard. 2015 [7]. |
| Month 5 | HIVneg_TB 0.000; HIVneg 0.015; HIVpos_TB 0.064; HIVpos 0.000; ART_TB 0.000; ART 0.000 | Fixed | Curve estimated from XTEND trial. Assume that the behaviour from out of care remains the same. Churchyard. 2015 [7]. |
| Month 6 | HIVneg_TB 0.000; HIVneg 0.010; HIVpos_TB 0.060; HIVpos 0.000; ART_TB 0.000; ART 0.000 | Fixed | Curve estimated from XTEND trial. Assume that the behaviour from out of care remains the same. Churchyard. 2015 [7]. |
| Probability of starting MDR-TB treatment if diagnosed with MDR-TB | HIVneg 0.025; HIVpos 0.019; ART 0.000 | Dirichlet | Estimated from XTEND trial. Churchyard. 2015 [7]. |
| *Treatment, transition probabilities* | | | |
| Probability of drug sensitive TB regimen started if TB treatment started | HIVneg 0.952; HIVpos 0.969; ART_TB 0.834 | Dirichlet | Estimated from XTEND trial. Churchyard. 2015 [7]. |
| Probability of MDR-TB regimen started if TB treatment started, mean (standard deviation) | HIVneg 0.039 (0.208); HIVpos 0.023 (0.002); ART 0.000 (0.000); | Dirichlet | Estimated from XTEND trial. Churchyard. 2015 [7]. |
| *Disease progression, transition probabilities* | | | |
| Average life expectancy at birth, South Africa | 63 years | Fixed | From the rapid mortality surveillance report 2014. Assumes that HIVpos patients who are on ART when they enter the model would have the same life expectancy as the general population (varied in the sensitivity analysis). HIV specific mortality considered in model through probabilities. Dorrington. 2015 [34]. Years of life remaining at death is estimated from the difference between current age in model (mean age of cohort + time in model) and the average life expectancy at birth. |
| All-cause mortality in those without TB, monthly, mean (standard deviation) | HIVneg 0.001 (0.0005); HIVpos 0.002 (0.000); ART 0.001 (0.001) | Dirichlet | From Statistics South Africa report (P0309.3), mortality and causes of death in South Africa: findings from death notification [35]. |
| Standardised mortality ratio for all-cause mortality in patients post-TB treatment | 3.76 | Fixed | Increased all-cause mortality in those with a previous episode of TB [36]. Estimated as part of a systematic review and meta-analysis. |
| Monthly mortality if living with TB, not currently receiving treatment, mean (standard deviation) | HIVneg 0.018 (0.020); HIVpos 0.132 (0.005); ART 0.039 (0.005) | Changes over time | Based on Tiemersma. 2011 [37]. Used half-cycle correction to adjust for earlier movement into treatment in month 1 of the model. |
| Monthly mortality on treatment for those with TB, mean (standard deviation) | HIVneg 0.002 (0.001); HIVpos 0.046 (0.002); ART 0.006 (0.003) | Changes over time | Andrews 2012 [38]. Mohr 2015 [39]. Monthly mortality reduction due to TB treatments added as distribution over time, where mortality reduces to 10% of the mortality of those with TB not on treatment at month 5 on treatment. Based on comparison with mortality on treatment observed in the XTEND trial. Churchyard 2015 [7]. |
| Disability weights, mean (standard deviation) The disability weight is a factor reflecting the severity of disease. | HIVneg_TB 0.331 (0.057); HIVpos_TB 0.399 (0.070); HIVpos 0.221 (0.041); ART 0.053 (0.011); ART_TB 0.331 (0.057) | Beta | Salomon. 2015 [40]. Kastien-Hilka. 2017 [41]. Assuming that disability weights are not cumulative, thus those on ART with TB have the same disability weight as someone with TB disease only. |
| *Cost and resource use* | | | |
| Microscopy, mean (standard deviation) | $6.30 ($1.34) | Gamma | Cunnama 2016 [42]. |

*(Continued)*

**Table 1.** (Continued)

| Definition | Mean and stratification | Distribution | Comments. References are listed as name of first author, year (Reference). |
|---|---|---|---|
| Xpert, mean (standard deviation) | $16.90 ($6.10) | Gamma | Cunnama 2016 [42]. |
| Sputum liquid culture, mean (standard deviation) | $12.90 ($2.26) | Gamma | Cunnama 2016 [42]. |
| Digital radiograph, mean (standard deviation) | $15.17 ($7.74) | Gamma | Foster. Unpublished. |
| First-line drug sensitivity test, mean (standard deviation) | $20.30 ($7.28) | Gamma | Cunnama 2016 [42]. |
| Second-line drug sensitivity test, mean (standard deviation) | $25.10 ($20.22) | Gamma | Cunnama 2016 [42]. |
| Provider cost of clinic visit for initial diagnosis and monitoring | $8.63 | Fixed | Vassall 2017 [43]. |
| Provider cost of clinic visit for treatment | $3.89 | Fixed | Vassall 2017 [43]. |
| Patient cost of clinic visit | $2.90 | Fixed | Foster 2015 [44]. |
| Guardian cost per clinic visit | $10.04 | Fixed | Foster 2015 [44]. |
| Cost of caregiver per day | $0.69 | Fixed | Foster 2015 [44]. |
| Resource use along the diagnostic pathway | Detailed input available from S1 Text. | Gamma | Estimated by disease progression. Reported in Vassall 2017 [43]. Foster 2015 [44]. |
| Provider cost of drug sensitive TB treatment, episode | $192.99 | Fixed | Estimated based on patient movements through care observed in the trial. Reported in Vassall 2017 [43]. Foster 2015 [44]. |
| Provider cost of multi-drug resistant TB treatment, episode | $10 802.66 | Fixed | Estimated based on patient movements through care observed in the trial. Reported in Vassall 2017 [43]. Foster 2015 [44]. |
| Patient cost of drug sensitive TB treatment, episode | Cost of accessing care associated $459.16; Cost of illness $135.94 | Time-dependent functions | Foster 2015 [44]. |
| Patient cost of multi-drug resistant TB treatment, episode | Cost of accessing care associated $3 592.27; Cost of illness $2 442.03 | Time-dependent functions | Foster 2015 [44]. |

In the Table, a fixed distribution refers to a distribution one where no uncertainty interval is estimated in keeping with calibration practice in complex models. Furthermore, IQR = interquartile range; TB = tuberculosis; MDR-TB = multi-drug resistant tuberculosis; Xpert = Xpert MTB/RIF; HIVpos = individuals HIV positive not yet started on anti-retroviral therapy; HIVpos_TB = individuals HIV positive with tuberculosis; ART = individuals HIV positive started on anti-retroviral therapy; ART_TB = individuals HIV positive on anti-retroviral therapy with tuberculosis.

NP) to observe the additive effects of the scenarios. Investments were modelled by altering parameters at key stages in the patient pathway and how these will increase the count of utilisation that increases costs and affects outcomes. The cost of facilitating change through changing behaviour, which we refer to as the transaction cost is shown in Fig 3.

## Economic analyses

The cost-effectiveness of investment scenarios was estimated from the societal perspective, which includes provider and patient-incurred costs. Disability adjusted life years (DALYs) averted were estimated using model estimates of years of life lost (YLL) due to premature mortality and years lived with disability (YLD). YLL were estimated based on progression through the model, assuming an average life expectancy of 63 years and the mean age (38 years) of patients in the trial [7, 34]. Disability weights from the 2010 Global Burden of Disease study were attached to model states [52]. For people on ART with tuberculosis, we assumed the

**Table 2. Summary of the investment scenarios modelled.**

| | Investment | Model implementation | Parameter, events or resource changes | Assumptions |
|---|---|---|---|---|
| Reduction in initial LTFU (in Fig 1; decision-point C and E) | All patients with positive TB test results start treatment within one month of testing—simulating a point-of-care or a track-and-trace scenario with active follow-up of people with a positive TB test result. Synergies with investment in a community health worker programme. | ptxfpos = 1—pMort_m1 (stratified by HIV and TB status) | Probability of starting treatment after positive (in month 1), from: | Monthly conditional probabilities of starting treatment from 'out of care' were estimated from the trial in the base scenario (reported in Table 1). In this investment scenario, patients shift from moving to the 'out of care' state if not started on treatment within one month, to the treatment state immediately, thus probabilities of starting treatment from 'out of care' approximate zero. The relative proportions of those starting various treatment types is kept the same as observed in the trial. |
| | | The probability of starting treatment from a positive test result was the remainder of all patients in that state after those who would die in that month had been subtracted. The mortality rate was stratified by HIV and TB status. | HIVneg: 0.882 to 1; | |
| | | | HIVneg_TB: 0.882 to 1; | |
| | | | HIVpos: 0.802 to 1; | |
| | | | HIVpos_TB: 0.802 to 1; | |
| | | | ART: 0.944 to 1; | |
| | | | ART_TB: 0.944 to 1 | |
| Empirical treatment from negative test result (in Fig 1; decision-point A) | The ability of healthcare workers to correctly act based on continued clinical symptoms, on the same day as the results visit (by giving TB treatment to those with test negative TB expressed as the sensitivity and specificity of that decision). This was based on the behaviour estimated from the microscopy arm of the model calibration and was applied to behaviour after a negative Xpert test result. | pnegpathfeg = 0 | Probability of the **negative pathway after a negative test result**, from: | Given the differences in health care worker behaviour after a microscopy test compared to a Xpert test result observed in the XTEND trial, we use the transition probabilities estimated from the microscopy arm of the trial [50, 51]. |
| | | ptreatfneg = value estimated from reported behaviour in the control arm of the XTEND study [49], under the assumption that behaviour observed after the implementation will revert back to pre-implementation levels. | HIVpos: 0.027 to 0.000 | |
| | | Assumed that all have at least one visit to a public health clinic (and associated costs) after a negative test result for treatment initiation. | HIVpos_TB: 0.212 to 0.000 | |
| | | | ART: 0.037 to 0.000 | |
| | | | ART_TB: 0.217 to 0.000 | |
| | | | Probability of **starting treatment after a negative test result**, from: | |
| | | | HIVneg: 0.002 to 0.040 | |
| | | | HIVneg_TB:0.054 to 0.270 | |
| | | | HIVpos: 0.009 to 0.180 | |
| | | | HIVpos_TB: 0.072 to 0.360 | |
| | | | ART: 0.003 to 0.060 | |
| | | | ART_TB: 0.017 to 0.090 | |

(*Continued*)

**Table 2.** (Continued)

| | Investment | Model implementation | Parameter, events or resource changes | Assumptions |
|---|---|---|---|---|
| Improvements in the test-negative pathway (in Fig 1 decision-points B and D) | HIV-positive people with negative test results get further investigations (radiograph/culture) for TB, and a proportion are started on TB treatment, simulating additional investment in improving access to further diagnostic tests. | ptreatfneg = 0 | Probability of **starting treatment after negative test result** changes from: | Similar to the previous scenario, we model a healthcare worker behaviour change scenario based on the difference in observed behaviour between the microscopy and Xpert arms of the study. This scenario simulates a situation where there is an increase in the proportion of patients who receive further investigations after a negative test result. Therefore, we reduced all empirical treatment to 0 and all eligible patients received a radiograph as part of the negative pathway. |
| | | pnegpathfneg = 1 (stratified by HIV and TB status) | HIVneg: 0.002 to 0.000 | |
| | | treatfnegpath = 0.10 (no TB); 0.80 (with TB) | HIVneg_TB:0.054 to 0.000 | |
| | | The probability of starting treatment is shifted from following a negative test result to the decision to order further diagnostic tests. The probability of starting treatment after the negative pathway was 10% in those without TB, and 80% in those with TB. | HIVpos: 0.009 to 0.000 | |
| | | Assumed that every person will accumulate two visits to the public clinic during the negative pathway, and that each person getting further tests will get at least one radiograph. | HIVpos_TB: 0.072 to 0.000 | |
| | | | ART: 0.003 to 0.000 | |
| | | | ART_TB: 0.017 to 0.000 | |
| | | | Probability of the **negative pathway after a negative test result** change from: | |
| | | | HIVpos: 0.041 to 0.900 | |
| | | | HIVpos_TB: 0.041 to 0.900 | |
| | | | ART: 0.073 to 0.900 | |
| | | | ART_TB: 0.073 to 0.900 | |
| | | | Probability of **treatment from negative pathway** changes from: | |
| | | | HIVpos: 0.027 to 0.100 | |
| | | | HIVpos_TB: 0.212 to 0.800 | |
| | | | ART: 0.037 to 0.100 | |
| | | | ART_TB: 0.217 to 0.800 | |

In the Table, the individual characteristics of the patients are labelled as HIVneg for people who are HIV negative and don't have tuberculosis; HIVneg_TB for people who are HIV negative and have been diagnosed with tuberculosis; HIVpos for people who are HIV positive and don't have tuberculosis; HIVpos_TB for people who are HIV positive and have been diagnosed with tuberculosis; ART represents the individuals who are HIV positive, on anti-retroviral therapy and don't have tuberculosis; and ART_TB represents the individuals who are HIV positive, on anti-retroviral therapy and have been diagnosed with tuberculosis.

same disability weight as for those with tuberculosis who are HIV negative. Costs and outcomes were discounted at 3% per annum, and varied in the sensitivity analyses [53: 108–112].

Transaction costs are conceptualised as the value of resources that would support better decision-making between agents. These costs are incurred during each decision-making interaction along the patient pathway (represented by blue dots in Fig 1). Changes in the optimal investment strategy at a range of transaction costs are evaluated by plotting cost-effectiveness acceptability frontiers (Fig 3) [54]. The optimal investment option is defined as the strategy with the highest net monetary benefit at a given cost-effectiveness threshold and transaction cost level.

### Sensitivity and scenario analyses

The impact of model parameters changes on results was assessed through univariate sensitivity analysis. Probabilistic uncertainty analyses, simulating 100 000 samples, were used to assess the simultaneous effect of path and parameter uncertainty on the results [55].

Scenario analyses were used to explore how implementation may vary between contexts. Given the set of interactions governing decision-making in the care pathway, some of which would be harder to mediate through additional investment [56], an increase in the value of supporting investments would not lead to proportional, linear improvements in outcomes [57].

**Ethics statement.** The study was approved by the research ethics committees of the University of Cape Town (363/2011), University of the Witwatersrand (M110827), London School of Hygiene & Tropical Medicine (6041), and the World Health Organization (RPC462). Health department officials and facility managers provided permission to conduct the study in the selected facilities and written informed consent was obtained from respondents.

## Results

After parameterising the model with data from the trial, and validating to the observed rate of TB treatment started and other secondary outcomes, we found that in order to achieve a good fit of the model to the data, we needed to also consider the limitations of sputum-based tuberculosis diagnostic modalities [58]. Undiagnosed tuberculosis may be related to the site of infection (extra-pulmonary tuberculosis), and low bacillary load in the sample, as is common in advanced HIV disease. During the model calibration, we therefore also added a parameter to capture the prevalence of extra-pulmonary tuberculosis (EPTB), varied along with the positive predictive value (PPV) and negative predictive value (NPV) to identify the best model fit. The prevalence of TB in the cohort was estimated to be 13% (see S1 Text).

### Costs, effectiveness, and cost-effectiveness analyses of the investment scenarios

Table 3 presents the costs, effectiveness (deaths averted and DALYs averted), and cost-effectiveness of investment scenarios, compared with the base case of Xpert as observed during the trial. The uncertainty interval (UI) is shown in brackets. From the provider's perspective, the incremental cost-effectiveness ratios (ICERs) ranged between $17.42 and $39.70 per DALY averted. We estimated a provider cost of tuberculosis services of $89.66 (UI: $87 - $92) per symptomatic person tested using an Xpert-based diagnostic algorithm. The societal cost per person was estimated to be $169.94 (UI: $167 - $173).

Reducing iLTFU by starting all individuals who test positive on treatment increased the cost of treatment and patient cost of accessing care per patient by $2.76 and $8.25 respectively. This scenario reduced time-to-treatment but has a comparatively small effect on the total number of people starting treatment and on health outcomes. Assuming that 100% start treatment in month one shifts the time-to-treatment started curve to the left, starting people on treatment who would have never started as well as those who would have started within the next couple of months. Since TB treatment does not instantly reduce mortality for patients who have TB, the proportion of patients who start treatment in month one in the reduction in iLTFU investment option only increases by 12% in those HIV negative, 10% in those HIV negative with TB, 20% in the HIV positive group, 7% in those HIV positive with TB, 6% in those on ART, and 2% in those on ART with TB.

**Table 3. Costs (US$), outcomes and ICERs over three years (36 one-month cycles) in a cohort with an estimated TB prevalence of 13%.**

| Status quo and five investment scenarios | In cohort of 10 000, true TB treated (range) | TB service costs per symptomatic individual (US$) | | | | Outcomes per symptomatic individual | | | | ICERs: compared against the status quo | | | |
|---|---|---|---|---|---|---|---|---|---|---|---|---|---|
| | | Provider costs | | Societal costs | | DALYs and DALYs averted | | Deaths and Deaths averted | | Provider cost/ DALY averted (95% UI) | Societal cost/ DALY averted (95% UI) | Provider cost/ death averted (95% UI) | Societal cost/ death averted (95% UI) |
| | | Total (95% UI) | Incr change from base (%) | Total (95% UI) | Incr % change (range) | Total DALYs (95% UI) | Incr DALYs averted % change (range) | Total deaths (95% UI) | Incr deaths averted % change (range) | (95% UI) | (95% UI) | (95% UI) | (95% UI) |
| Xpert (status quo) | 940 | 89.66 | - - - | 169.94 | - - - | 4.72 | - - - | 0.133 | - - - | - - - | - - - | - - - | - - - |
| | (920; 960) | (87; 92) | | (167; 173) | | (4.6; 4.8) | | (0.129; 0.136) | | | | | |
| Xpert plus reduction in initial LTFU (iLTFU) | 1010 | 92.42 | 2.76 | 178.19 | 8.25 | 4.56 | 0.16 | 0.128 | 0.005 | 17.42 | 51.86 | 601.40 | 1790.50 |
| (C and E) | (990; 1030) | (90; 95) | 3% | (175; 181) | 5% | (4.4; 4.7) | 3% | (0.125; 0.132) | 4% | (2.2; 117.6) | (18.5; 271.0) | (75.1; 3806.7) | (644; 8774) |
| Xpert plus treatment from negative (TfN) | 1140 | 110.78 | 21.12 | 256.36 | 86.42 | 4.04 | 0.68 | 0.115 | 0.018 | 31.40 | 128.45 | 1180.00 | 4826.60 |
| (A and E) | (1120; 1160) | (109; 113) | 24% | (253; 260) | 51% | (3.9; 4.1) | 14% | (0.112; 0.118) | 14% | (24.6; 40.5) | (107.2; 157.0) | (905.9; 1567.3) | (3939.0; 6079.2) |
| Xpert plus reduction in initial LTFU, and treatment from negative (iLTFU_TfN) | 1210 | 113.55 | 23.89 | 264.60 | 94.66 | 3.88 | 0.84 | 0.110 | 0.023 | 28.73 | 113.82 | 1061.70 | 4205.70 |
| (A, C and E) | (1190; 1230) | (111; 116) | 27% | (261; 268) | 56% | (3.8; 4.0) | 18% | (0.107; 0.113) | 17% | (23.5; 35.3) | (98.3; 133.3) | (853.3; 1333.3) | (3569.1; 5035.1) |
| Xpert plus improvements in the negative pathway (NP) | 1420 | 141.01 | 51.35 | 278.87 | 108.93 | 3.42 | 1.30 | 0.096 | 0.037 | 39.70 | 84.19 | 1387.70 | 2943.10 |
| (B, D and E) | (1390; 1450) | (139; 143) | 57% | (274; 284) | 64% | (3.3; 3.5) | 28% | (0.093; 0.099) | 28% | (35.2; 44.9) | (75.0; 94.8) | (1225.6; 1576.9) | (2608.4; 3334.0) |
| Xpert plus reduction in initial LTFU, and improvements in the negative pathway (iLTFU_NP) | 1480 | 142.99 | 53.33 | 285.97 | 116.03 | 3.28 | 1.44 | 0.092 | 0.041 | 37.02 | 80.55 | 1292.97 | 2813.00 |
| (B, C, D and E) | (1460; 1510) | (141; 145) | 59% | (281; 291) | 68% | (3.2; 3.4) | 31% | (0.089; 0.094) | 31% | (33.3; 41.3) | (72.8; 89.4) | (1155.8; 1449.9) | (2527.7; 3139.4) |

In the Table, Incr is the incremental change in costs or effectiveness from the base case. The base case in this analysis which represents the current status quo, Xpert as observed in the intervention arm of the XTEND study; dominant: less costly and more effective; dominated: more costly and less effective; The 95% uncertainty interval (UI) is shown in parentheses; ICER: Incremental cost-effectiveness ratio; DALYs: Disability Adjusted Life Years. In the scenario column, the capital letters refer to the decision points upon which the investment scenario acts, as shown in Fig 1.

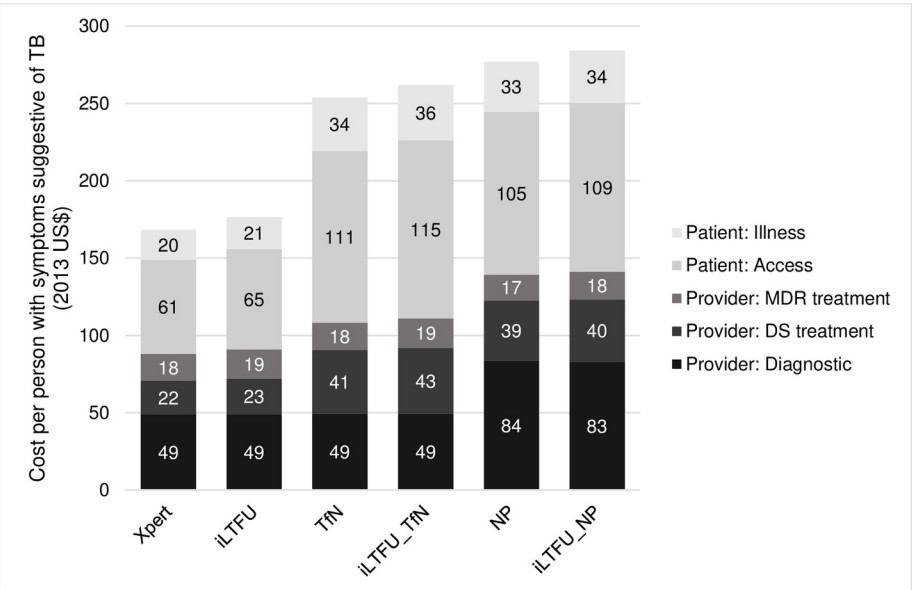

**Fig 2. Societal service-level costs (US$) per symptomatic person per episode.** In the Figure, the cost of accessing care (Access) includes out of pocket and time costs incurred by patients and caregivers when accessing care; the cost of illness (Illness) includes the cost of caregiver's time, the cost of patient's time when unable to work as well as loan interest, assets sold and the cost of nutritional supplements. Xpert referes to the Xpert baseline; iLTFU (Xpert plus iLTFU) = additional investment to reduce pre-treatment loss-to-follow-up; TfN (Xpert plus TfN) = supporting clinical diagnosis of tuberculosis after a negative test results; Np (Xpert plus NP) = improving access to further tuberculosis diagnostic tests following a negative test result.

Supporting same-day clinical diagnosis of TB after a negative tuberculosis test result increases the cost of the TB service per symptomatic person per episode by $21.12 due to the increase in patients started on TB treatment, with likewise an increase in societal costs associated with accessing treatment of $86.42 per patient (Fig 2).

In contrast, improving access to further diagnostic tests following a negative test result (negative pathway) increases diagnostic costs by $35 per patient due to the follow-on tests ordered, with an increase in the cost of treatment (Fig 2). This scenario increases the patient costs associated with accessing care (from $61 to $105 per patient) as patients make multiple visits for follow-on diagnostic tests and results. In addition, delays in starting treatment increase the cost of illness due to a loss of time and income.

For people with a negative Xpert test result, our analysis suggest that further testing (negative pathway), as conceptualised here, may be more effective at reducing mortality than empirical treatment; however the provider costs per symptomatic individual are considerably higher at $141.01 ($139 - $143) versus $110.78 ($109–$113) in the negative pathway compared against treatment started following a clinical diagnosis. Similarly, societal costs are higher due to increased diagnostic visits and delays in starting treatment increase the cost of illness which is based on caregiver's time as well as patient's time unable to work.

## Transaction cost analysis

Using a cost-effectiveness threshold that reflects recent decisions adopted by the South African government (revealed willingness-to-pay) [59], we find that investments of up to $601 per symptomatic individual would be cost-effective. It is therefore likely that considerable

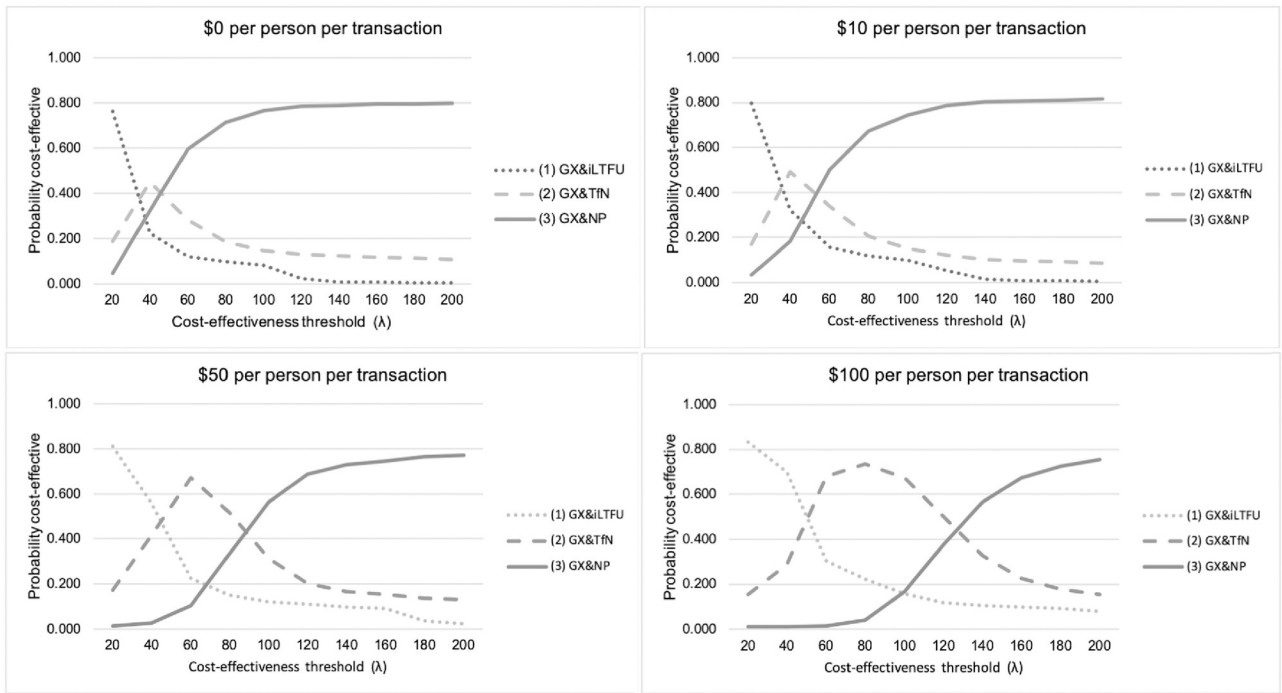

**Fig 3. Provider cost-effectiveness acceptability frontiers (CEAF) at various levels of transaction costs.** Where iLTFU refers to Xpert plus a reduction in initial loss to follow up scenario; TfN refers to the scenario modelling Xpert plus treatment from negative; Np refers to Xpert plus improvements in the negative pathway. The cost-effectiveness acceptability frontier (CEAF) expressing the uncertainty around the cost-effectiveness of investments, by showing which strategy is economically preferred at a range of cost-effectiveness thresholds (on the x-axis). The base case scenario for each of these comparisons is Xpert MTB/RIF, as observed in the XTEND trial. The graph is a plot of the proportion of individual runs that would be cost-effective for each intervention (y-axis) while restricting the options to only those that would be the most cost-effective (optimal) investment for at least one individual, against a range of cost-effectiveness thresholds (x-axis). As the threshold increase, the preferred option changes, the switch point being where the incremental cost-effectiveness ratio (ICER) value increases beyond the threshold [62]. The analysis is repeated at a range of transaction costs per transaction, thereby varying the costs needed to be invested to facilitate systems level change in line with the investment strategy.

investments in strengthening supportive systems around TB diagnosis in South Africa would be value for money.

Fig 3 presents the cost-effectiveness acceptability frontiers, which show the optimal provider investments at a range of transaction costs and cost-effectiveness thresholds. As explained, transaction costs are modelled per transaction, and are conceptualized as the resources needed to improve decision-making within each investment scenario. Assuming no transaction costs, investing in reducing initial loss-to-follow up was the optimal investment if the cost-effectiveness threshold was below $30/ DALY averted, but at higher thresholds, the negative pathway was the optimal investment. As the investment cost per person per transaction increased, empirical treatment became the optimal investment compared to the negative pathway at lower cost-effectiveness thresholds. This is driven by a reduction in healthcare visits when patients are started on treatment empirically.

## Sensitivity analyses

Detailed results of the univariate sensitivity analyses are included in S1 Text, summarised in Fig 4.

Intervention provider costs is dependent on population characteristics. For example, if much of the population is HIV-positive not taking antiretroviral therapy, a higher proportion with tuberculosis would test negative, leading to higher costs, though this would be mediated

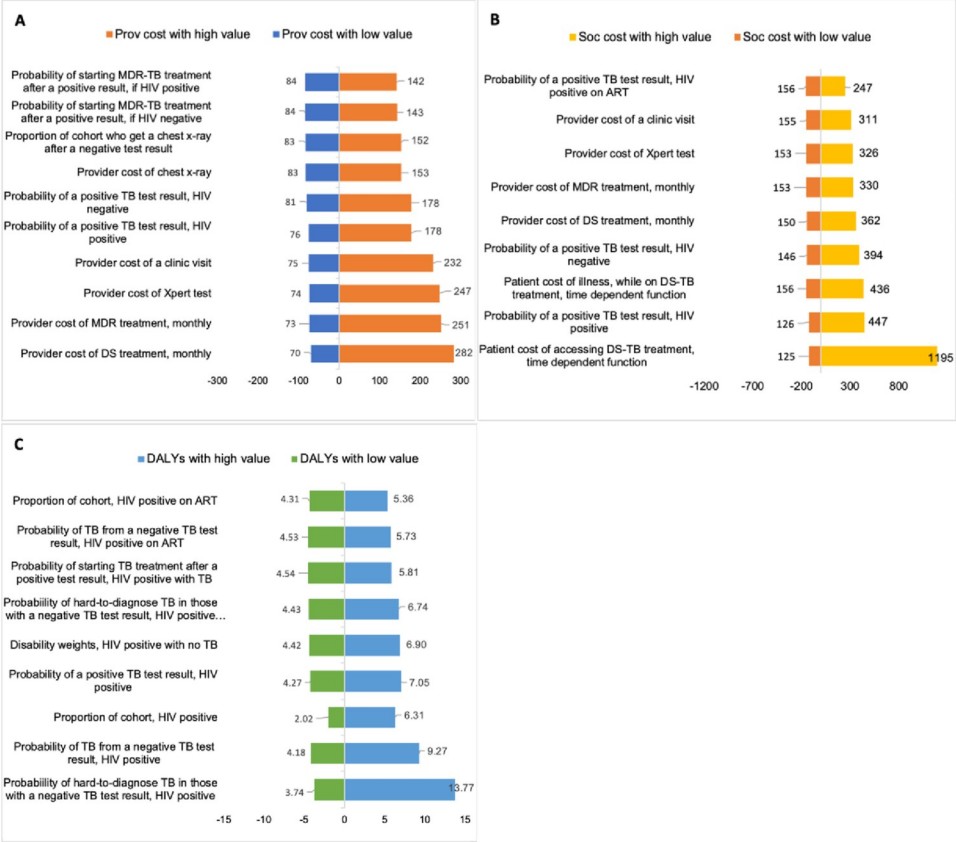

**Fig 4.** (A-C) Results from the univariate sensitivity analyses, showing the ten parameters with the greatest influence on the (A) provider cost, (B) the societal costs, and the (C) effectiveness (DALYs) of the base case (Xpert). The full results for these analyses are presented S1 Text. In each one-way analysis, one parameter was varied by a factor of 10 from the mean to produce the low and high estimates, with all other parameters kept constant. Where DALYs are disability adjusted life years; Prov refers provider; and Soc is societal. DS treatment is drug-sensitive treatment. MDR refers to multi-drug resistant tuberculosis.

by the expansion of universal access to antiretroviral therapy. Similarly, the effectiveness of these investments is sensitive to the health-seeking behaviours of patients and health system characteristics, specifically whether patients return for their results, the availability of chest radiographs and whether treatment is started after further diagnostic tests. The prevalence of multi-drug resistant tuberculosis and the cost of multi-drug resistant tuberculosis treatment was an important driver of costs and effectiveness of the overall results.

## Discussion

Our analyses build on a global body of work evaluating the use of Xpert-based diagnostic pathways [4, 10, 15, 60–63] by presenting the cost-effectiveness of complementary investments to strengthen the diagnostic pathway [64]. We explored how investments in health system to support the patient pathway may affect the resource use and outcomes associated with tuberculosis diagnostics. Our findings suggest that it is unlikely that a single investment or technology would dramatically improve the outcomes of symptomatic patients receiving a tuberculosis diagnostic test; instead our results suggest that investments in various parts of the care pathway could generate additional benefits, and, based on the transaction cost analysis, we show that relatively high levels of investment in health systems strengthening may be cost-effective.

When comparing across the care pathway (Table 2), our analysis finds that in a symptomatic cohort with 13% prevalence of tuberculosis, only minor reductions in mortality can be achieved by improving initial pre-treatment loss to follow up, while much larger benefits can be achieved by improving access to further tests after a negative tuberculosis test. This may be explained, in part, by the higher mortality rates observed in people who are HIV-positive with an initial negative tuberculosis test result.

## Potential drivers of investment value

While the Xpert assay automates diagnostic processes and provides test results within two hours, in South Africa, Xpert machines were placed at laboratories with results delivered to health facilities in two days. Therefore, despite Xpert implementation reducing the turn-around time of results, follow-up clinic visits by patients were still required [42]. The need to improve the linkage of patients with their results has been highlighted as an important component of better tuberculosis diagnosis, however our analysis suggests that comparatively low gains in terms of mortality reduction would be achieved in such an investment scenarios. This may be explained by lower mortality rates in those with positive sputum test results, and that people-living-with-HIV who have high rates of tuberculosis-associated mortality are less likely to have a positive sputum test result. While the mortality reduction is likely to be modest, those with positive tuberculosis test results are potentially transmitting tuberculosis in communities, increasing the future burden of need at a population level [65]. These results are somewhat supported by findings from studies that highlighted the challenges of point-of-care Xpert testing at facilities in urban settings [66] and benefits in rural communities [67].

Clinical decision-making after a negative test result is important in understanding the cost-effectiveness of new tuberculosis diagnostics, suggesting that greater awareness of tuberculosis symptoms among health care workers may improve outcomes and be a cost-effective intervention [10, 68, 69]. In Uganda, Hermans et al. (2017) found that tuberculosis treatment was initiated based on clinical symptoms in 17% of patients for whom an Xpert test was requested [50]. In South Africa, an evaluation of tuberculosis programmatic data found that there was a decline in the use of empirical tuberculosis treatment from 42% to 27% following the introduction of Xpert [51]. It is possible that the introduction of Xpert did not significantly reduce tuberculosis-associated mortality due, in part, to a reduction in action, including follow-on tests, after a negative test result [49]. Access to further tests such as chest radiography and mycobacterial culture of sputum after a negative result is dependent on the availability of chest radiography in close proximity to the health facility, how healthcare workers use these tests, as well as access barriers to patients [70–72]. Our analysis suggests that assumptions of how quickly tuberculosis treatment reduces mortality rates is a key determinant of the effectiveness of this strategy.

## Investing in health systems strengthening

While it is not possible to say whether an investment scenario is cost-effective without consensus on a cost-effectiveness threshold in South Africa, we find that investing in strengthening health systems to support the tuberculosis diagnostic algorithm is likely to be a high value investment. The outcomes of these investments are also likely to influence other disease programs and sectors [73]. We do not include these spill over benefits or costs in our analysis, and thus our estimates are conservative. Empirical work has highlighted the importance of going beyond investing in assets and technology to invest in developing agency and governance (the software capacities of health systems) [74]. Those investments are highly contextual and

difficult to cost, so while our approach highlights to decision makers the resource envelopes required, more work is needed to develop and iteratively assess context-specific investment strategies. In-depth qualitative work to understand the barriers and facilitators of health care workers' implementation of diagnostic guidelines would fill some of this gap.

The following limitations should be considered when interpreting our findings. Firstly, we did not model the effect of the various scenarios on the tuberculosis epidemic at a population level. While the implementation of Xpert primarily resulted in an increased identification of smear-negative tuberculosis, currently thought not to be a major driver of transmission, not including transmission in the analysis is likely to underestimate the relative benefit of reducing pre-treatment LTFU at a population level [65, 75, 76]. Secondly, while we are modelling scenarios and benefits in a nuanced way, the relationship between health system structures, health care worker -, and patient behaviour is complex and while one can observe patterns, predictions will be limited by our understanding of the mechanisms driving these patterns. Thirdly, any investment in the health system will be likely to have an impact on other associated services (externalities), the benefits of which we did not include in our analysis [42]. Lastly, while our model includes a pathway for patients to initiate multi-drug resistant tuberculosis treatment if diagnosed, and incur the associated costs, we do not attempt to estimate the true prevalence of multi-drug resistant tuberculosis or what effect the investments may have on the epidemic. In the analysis of the negative pathway, therefore, the model may be underestimating the effect of incorrectly starting an individual on drug-sensitive tuberculosis. Studies following the roll-out of Xpert have found that barriers to initiating multi-drug resistant tuberculosis persisted and that the time-to-appropriate-treatment was only slightly reduced [8, 77].

In conclusion, our findings suggest that within the context of a high tuberculosis prevalence setting, with a well-developed laboratory infrastructure, the implementation of new tuberculosis diagnostics should be accompanied by additional investments in the health system. Current international policy is to substantially expand and intensify tuberculosis detection, yet if this is not accompanied by investments to support decision-making after a negative test result, it is unlikely that these efforts alone will modify the tuberculosis epidemic.

## Supporting information

**S1 Text. Technical appendix.**
(DOCX)

**S1 Data. Parameter list accompanying manuscript Foster et al. strengthening health systems to improve the value of tuberculosis diagnostics in high-burden settings: A cost and cost-effectiveness analysis.**
(XLSX)

## Acknowledgments

This study is part of the "Xpert for TB: Evaluating a New Diagnostic" (XTEND) project. This work would not have been possible without the many generous contributions of the study team, the respondents and health facility staff. In particular, the authors would like to thank Professor Anna Vassall for her contributions to this work. We are also grateful to the staff of the UCT Health Economics Unit; the Health Policy and Systems Division; as well as the UCT School of Public Health and Family Medicine for their insights and contributions to discussions of the work presented here.

## Author Contributions

**Conceptualization:** Nicola Foster, Lucy Cunnama, Alison D. Grant, Susan Cleary.

**Data curation:** Nicola Foster, Lucy Cunnama, Kerrigan McCarthy, Lebogang Ramma, Mariana Siapka, Edina Sinanovic, Katherine Fielding.

**Formal analysis:** Nicola Foster, Mariana Siapka, Susan Cleary.

**Funding acquisition:** Nicola Foster, Edina Sinanovic, Gavin Churchyard.

**Investigation:** Nicola Foster, Lucy Cunnama, Kerrigan McCarthy, Lebogang Ramma, Gavin Churchyard, Katherine Fielding, Alison D. Grant, Susan Cleary.

**Methodology:** Nicola Foster, Susan Cleary.

**Project administration:** Nicola Foster.

**Resources:** Kerrigan McCarthy, Lebogang Ramma, Mariana Siapka, Gavin Churchyard.

**Supervision:** Edina Sinanovic, Alison D. Grant, Susan Cleary.

**Writing – original draft:** Nicola Foster.

**Writing – review & editing:** Lucy Cunnama, Kerrigan McCarthy, Lebogang Ramma, Mariana Siapka, Edina Sinanovic, Gavin Churchyard, Katherine Fielding, Alison D. Grant, Susan Cleary.

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
