## [Decision Letter · Decision Letter 0]

2 Sep 2020

PONE-D-20-18039

Strengthening health systems to improve the value of Tuberculosis diagnostics in South Africa: a cost and cost-effectiveness analysis.

PLOS ONE

Dear Dr. Foster,

Thank you for submitting your manuscript to PLOS ONE. After careful consideration, we feel that it has merit but does not fully meet PLOS ONE’s publication criteria as it currently stands. Therefore, we invite you to submit a revised version of the manuscript that addresses the points raised during the review process.

Please submit your revised manuscript. If you will need more time than this to complete your revisions, please reply to this message or contact the journal office at plosone@plos.org. Please include the following items when submitting your revised manuscript:

We look forward to receiving your revised manuscript.

Kind regards,

Frederick Quinn

Academic Editor

PLOS ONE

Journal Requirements:

2. Your ethics statement must appear in the Methods section of your manuscript. If your ethics statement is written in any section besides the Methods, please move it to the Methods section and delete it from any other section. Please also ensure that your ethics statement is included in your manuscript, as the ethics section of your online submission will not be published alongside your manuscript.

3. Please ensure that you refer to Figure 2 in your text as, if accepted, production will need this reference to link the reader to the figure.

Reviewers' comments:

Reviewer's Responses to Questions

**Comments to the Author**

1. Is the manuscript technically sound, and do the data support the conclusions?

Reviewer #1: Partly

Reviewer #2: Yes

2. Has the statistical analysis been performed appropriately and rigorously? 

Reviewer #1: Yes

Reviewer #2: Yes

3. Have the authors made all data underlying the findings in their manuscript fully available?

Reviewer #1: Yes

Reviewer #2: Yes

4. Is the manuscript presented in an intelligible fashion and written in standard English?

Reviewer #1: No

Reviewer #2: Yes

5. Review Comments to the Author

Reviewer #1: The authors seek to evaluate where the bottleneck is when efforts are put to improve health outcome for TB community. Walking through the journey from TB diagnostics to timely and effective treatment, the authors identified that improving decision making for TB patients who received negative results from the first-round diagnostic tests. This study has its merits in terms of significance for TB community. Please address the following comments of this reviewer:

1. The introduction contains ambiguity. For instance, in Page 11 Line 20-21, you cited a reference (Reference 10) and stated that Xpert implementation is cost- and effect-neutral. However, in that reference, the conclusion was proper ineffective implementation of Xpert was the limiting factor to improve outcome. The statement in the current manuscript could cause confusion whether Xpert technology itself is not effective or there lacks infrastructure for full utilization of this technology.

2. If the false-negative rate of Xpert is not significant, or the portion of patients with both TB and HIV is not significant, how do you justify your conclusion. In addition, It would be helpful to your conclusion if diagnostic capability of Xpert were briefly introduced to exclude this as a variable in your model.

3. Please provide a clear definition for cost-effectiveness.

4. Please fix all language issues throughout the manuscript.

Reviewer #2: Manuscript Review: PONE-D-20-18039

Strengthening health systems to improve the value of Tuberculosis diagnostics in South Africa: a cost and cost-effectiveness analysis

Nicola Foster, Lucy Cunnama, Kerrigan McCarthy, Lebogang Ramma, Mariana Siapka, Edina Sinanovic, Gavin Churchyard, Katherine Fielding, Alison D Grant, Susan Cleary

Key Results:

In the manuscript the authors attempt to determine the cost effectiveness of implementing further investments that can complement the use of Xpert-MTB-RIF testing in South Africa by building a mathematical model and incorporating variables. The model was used to identify which investment variable would increase cost effectiveness and improve Tuberculosis diagnostics to alleviate disease burden. In addition to monetary values, the authors also included deaths averted and disability-adjusted-life-years (DALYs) in their analysis. After their analysis, the authors found that investing in several different aspects of tuberculosis care is more cost effective than a single investment. Data showed that in symptomatic patients, costs were only marginally reduced by improving loss to follow-up and a greater good can be achieved by improving access to further testing. Overall, the authors suggest that an investment in the strengthening of the healthcare system’s tuberculosis diagnostics may be an important place to concentrate funds and not just investments in detection.

Validity:

Based on the methodology used in this study, I find the manuscript to be valid.

Originality and Significance:

I find the data presented in this study to be original and significant. Although what the authors suggest will be difficult to do and will face a lot of red tape, in theory, it is good data to have available.

Data and Methodology:

This manuscript presented a straightforward data analysis based on data collected through analysis of a tuberculosis cohort in South Africa with a 13% positive rate. The methodology presented is commonly used and appropriate.

Appropriate us of Statistics:

Appropriate statistics were used throughout the study. Proper statistics were conducted, and variables were adjusted for when needed.

Conclusions:

Based on the statistics presented in the study, the conclusions appear to be valid and reliable.

Suggested Improvements:

I did not find much improvement needed with the manuscript as a whole. The only issue was the sections where there was an error in the references. For example, page 20 lines 15 and 19. These need to be addressed.

References:

The references are valid.

Clarity and context:

The abstract, introduction and conclusions are clear, concise and appropriate.

Scope of expertise:

This manuscript is within the scope of my expertise.

6. PLOS authors have the option to publish the peer review history of their article (what does this mean?). If published, this will include your full peer review and any attached files.

Reviewer #1: No

Reviewer #2: No

---

## [Author Response · Author response to Decision Letter 0]

23 Oct 2020

PONE-D-20-18039

Strengthening health systems to improve the value of Tuberculosis diagnostics in South Africa: a cost and cost-effectiveness analysis.

Nicola Foster, Lucy Cunnama, Kerrigan McCarthy, Lebogang Ramma, Mariana Siapka, Edina Sinanovic, Gavin Churchyard, Katherine Fielding, Alison D Grant, Susan Cleary

Response to reviewers

Thank you for submitting your manuscript to PLOS ONE. After careful consideration, we feel that it has merit but does not fully meet PLOS ONE’s publication criteria as it currently stands. Therefore, we invite you to submit a revised version of the manuscript that addresses the points raised during the review process.

Please submit your revised manuscript. If you will need more time than this to complete your revisions, please reply to this message or contact the journal office at plosone@plos.org. Please include the following items when submitting your revised manuscript:

Thank you for reviews of the manuscript. Our revised submission includes the following documents:

• response_to_reviewers.docx

• revised_manuscript.docx (with track changes)

• manuscript.docx (without track changes)

• figures

• S1_Text.docx

• S2_Data.xlsx

Editor:

 Files have been renamed in accordance with the style requirements.

The manuscript style for main body and title page has been amended in the manuscript.

2. Your ethics statement must appear in the Methods section of your manuscript. If your ethics statement is written in any section besides the Methods, please move it to the Methods section and delete it from any other section. Please also ensure that your ethics statement is included in your manuscript, as the ethics section of your online submission will not be published alongside your manuscript.

 Ethics statement has now been placed at the end of the Methods section (pg 16, ln 4).

3. Please ensure that you refer to Figure 2 in your text as, if accepted, production will need this reference to link the reader to the figure.

 The within text reference links to Figure 2 have been fixed. (pg 19, ln 15). 

Reviewer #1: 

The authors seek to evaluate where the bottleneck is when efforts are put to improve health outcome for TB community. Walking through the journey from TB diagnostics to timely and effective treatment, the authors identified that improving decision making for TB patients who received negative results from the first-round diagnostic tests. This study has its merits in terms of significance for TB community. Please address the following comments of this reviewer: 

1. The introduction contains ambiguity. For instance, in Page 11 Line 20-21, you cited a reference (Reference 10) and stated that Xpert implementation is cost- and effect-neutral. However, in that reference, the conclusion was proper ineffective implementation of Xpert was the limiting factor to improve outcome. The statement in the current manuscript could cause confusion whether Xpert technology itself is not effective or there lacks infrastructure for full utilization of this technology. Removed ambiguity on page 5 by correcting the references to make it clearer that the statement “The study concluded that implementation constraints may have mediated the impact of Xpert under programmatic conditions. (7;10)” is a conclusion from the same study.

2. If the false-negative rate of Xpert is not significant, or the portion of patients with both TB and HIV is not significant, how do you justify your conclusion. In addition, it would be helpful to your conclusion if diagnostic capability of Xpert were briefly introduced to exclude this as a variable in your model. The relationship between the diagnostic capability of Xpert and the study population is discussed in more detail in the supplementary appendix (S1 Text pages 12 and 13), especially how this is included in the model and in the sensitivity analysis.

3. Please provide a clear definition for cost-effectiveness. Definition added on page 7, line 4: “Cost-effectiveness analysis is a method for examining the change in costs and the change in health outcomes of a given intervention.”

4. Please fix all language issues throughout the manuscript.

 The manuscript has been proof-read including by co-authors for whom English is their mother-tongue.

Reviewer #2: 

Manuscript Review: PONE-D-20-18039

Key Results:

In the manuscript the authors attempt to determine the cost effectiveness of implementing further investments that can complement the use of Xpert-MTB-RIF testing in South Africa by building a mathematical model and incorporating variables. The model was used to identify which investment variable would increase cost effectiveness and improve Tuberculosis diagnostics to alleviate disease burden. In addition to monetary values, the authors also included deaths averted and disability-adjusted-life-years (DALYs) in their analysis. After their analysis, the authors found that investing in several different aspects of tuberculosis care is more cost effective than a single investment. Data showed that in symptomatic patients, costs were only marginally reduced by improving loss to follow-up and a greater good can be achieved by improving access to further testing. Overall, the authors suggest that an investment in the strengthening of the healthcare system’s tuberculosis diagnostics may be an important place to concentrate funds and not just investments in detection.

Validity:

Based on the methodology used in this study, I find the manuscript to be valid.

Originality and Significance:

I find the data presented in this study to be original and significant. Although what the authors suggest will be difficult to do and will face a lot of red tape, in theory, it is good data to have available.

Data and Methodology:

This manuscript presented a straightforward data analysis based on data collected through analysis of a tuberculosis cohort in South Africa with a 13% positive rate. The methodology presented is commonly used and appropriate.

Appropriate us of Statistics:

Appropriate statistics were used throughout the study. Proper statistics were conducted, and variables were adjusted for when needed.

Conclusions:

Based on the statistics presented in the study, the conclusions appear to be valid and reliable.

Suggested Improvements:

I did not find much improvement needed with the manuscript as a whole. The only issue was the sections where there was an error in the references. For example, page 20 lines 15 and 19. These need to be addressed.

References:

The references are valid.

Clarity and context:

The abstract, introduction and conclusions are clear, concise and appropriate.

Scope of expertise:

This manuscript is within the scope of my expertise.

 Corrected the error in links to Figure 2 (pg 19, ln 15).

---

## [Decision Letter · Decision Letter 1]

24 Nov 2020

PONE-D-20-18039R1

Strengthening health systems to improve the value of Tuberculosis diagnostics in South Africa: a cost and cost-effectiveness analysis.

PLOS ONE

Dear Dr. Foster,

Thank you for submitting your manuscript to PLOS ONE. After careful consideration, we feel that it has merit but does not fully meet PLOS ONE’s publication criteria as it currently stands. Therefore, we invite you to submit a revised version of the manuscript that addresses the points raised during the review process.

Please submit your revised manuscript. If you will need more time than this to complete your revisions, please reply to this message or contact the journal office at plosone@plos.org. Please include the following items when submitting your revised manuscript:

We look forward to receiving your revised manuscript.

Kind regards,

Frederick Quinn

Academic Editor

PLOS ONE

Reviewers' comments:

Reviewer's Responses to Questions

**Comments to the Author**

1. If the authors have adequately addressed your comments raised in a previous round of review and you feel that this manuscript is now acceptable for publication, you may indicate that here to bypass the “Comments to the Author” section, enter your conflict of interest statement in the “Confidential to Editor” section, and submit your "Accept" recommendation.

Reviewer #3: (No Response)

Reviewer #4: (No Response)

2. Is the manuscript technically sound, and do the data support the conclusions?

Reviewer #3: Yes

Reviewer #4: Partly

3. Has the statistical analysis been performed appropriately and rigorously? 

Reviewer #3: Yes

Reviewer #4: I Don't Know

4. Have the authors made all data underlying the findings in their manuscript fully available?

Reviewer #3: Yes

Reviewer #4: Yes

5. Is the manuscript presented in an intelligible fashion and written in standard English?

Reviewer #3: Yes

Reviewer #4: Yes

6. Review Comments to the Author

Reviewer #3: The authors present a model of data from the XTEND study to look at the impact of various inputs into the tuberculosis program on the effect of Xpert on TB outcomes. The authors conclude that Xpert remains fairly neutral with respect to its impact. I question the focus on outcomes like mortality, as opposed to program outcomes which would include transmission. The model is already highly complex, but transmission seems like it would be an important consideration in this analysis. If mortality is relatively low, as it is for HIV negative patients, we won't see much of an effect. The authors state that improving diagnostic testing for patients with negative Xpert results would have an impact, reflecting higher mortality in HIV positive patients. However, it wasn't clear that results were quite sensitive to HIV rates in the population. I also question complete lack of exploration of MDRTB rates and outcomes, and very low treatment rates, as Xpert should impact management of those cases. There are aspects of this analysis that would be very difficult to quantify, like resources that would support physician decision making. We don't have an idea of any downstream effects of the intervention, which limits the impact of this study.

Reviewer #4: The study is fascinating and concerns a typical example of the importance of implementation science, or translational science. In this case, it has been shown in several studies that Xpert is more sensitive and specific than the microscope in diagnosing tuberculosis. A more specific and sensitive tool allows identifying more cases and starting treatment earlier, with a health benefit and also an economic benefit. This is suggested, for example, in the study conducted by Orlando et al. (Orlando S, Triulzi I, Ciccacci F, et al. . Delayed diagnosis and treatment of tuberculosis in HIV + patients in Mozambique: A cost-effectiveness analysis of screening protocols based on four symptom screening, smear microscopy, urine LAM test and Xpert MTB / RIF. PLoS One. 2018; 13 (7): e0200523 .)

However, this evidence derived from trials or simulations is not confirmed in a pragmatic trial, that explore the use of the apparently best technology in the real world. The authors rightly hypothesize that the problem does not lie in the instrument (Xpert) but in its use by physicians and its ineffective inclusion within the diagnostic-therapeutic path.

Therefore it is exciting to evaluate how an additional investment in these aspects can modify the effectiveness and cost-effectiveness of the most advanced technology compared to the status-quo (the microscope).

However, although the problem is very clear, in the course of the analysis, the fundamental steps concerning the issue in question are not clear, at least to the reader.

It is said that the question is relational and studies in the field of sociology are mentioned, such as reference 23. In this case, it would have been necessary to approach the problem from a sociological point of view. Therefore a qualitative analysis that analyzed in-depth the barriers to correct use of Xpert would be recommended.

Instead, the issue is addressed from an economic point of view, assuming that the ineffective use of technology depends on some lack of investment.

At this point, it would have been necessary to describe better what these investments are and what is their cost is. In the text, you never find this description nor an analysis of the costs of the 3 investments. Table 2 briefly describes the investments but does not explain in detail how these investments generate additional costs, what additional resources they need, and the cost of these resources.

In practice, all cost analysis is deferred to other studies cited, especially Foster 2015 and Vassall 2017. But if this is the fundamental point of this analysis, and this aspect should be reported also in this study.

Also from the point of view of effectiveness, it is not clear the mechanism through which these investments increase effectiveness, or rather allow to avert DALYs. Presumably through a reduction in mortality linked to a earlier identification of positive subjects, and a quicker start of treatment. However these passages are not reported in the methodology and in the discussion, where I expected them to be the most important aspect to discuss, together to what was said above about costs.

In essence, the mathematical model is a bit of a black box in which a higher cost generate a health benefit, but it is not clear, at least if we stick to the text, how these costs and benefits are generated.

I suggest making these aspects clear in the text and not only in the tables. Also, cost analysis should be described more in detail, and possibly discuss how other studies have estimated the cost of these interventions (if any).

Figures are not explained in the text. For example, figure 1, is not clear how it fits in the discourse and how to interpret it.

Finally, the conversion rate in USD should consider Power Purchasing Parity for cost generated in South Africa, such as costs incurred by patients.

In the table 3, all alternatives are compared with the base case scenario (Microscopy), but I would have reached them in order of increased effectiveness. In this way, if I'm not wrong, some alternative will be excluded becuse they are dominated by others with lower cost and higher effectiveness. I could be wrong about this, but with this presentation of results is not possible for the reader to clearly identify dominated or extendedly dominated alternatives.

7. PLOS authors have the option to publish the peer review history of their article (what does this mean?). If published, this will include your full peer review and any attached files.

Reviewer #3: No

Reviewer #4: No

---

## [Author Response · Author response to Decision Letter 1]

16 Apr 2021

Author responses to reviewers

PONE-D-20-18039

6 April 2021

Thank you to the editors and reviewers for your thoughtful comments on the manuscript, “Strengthening health systems to improve the value of tuberculosis diagnostics in South Africa: a cost and cost-effectiveness analysis”.

Reviewer #3: 

The authors present a model of data from the XTEND study to look at the impact of various inputs into the tuberculosis program on the effect of Xpert on TB outcomes. The authors conclude that Xpert remains fairly neutral with respect to its impact. 

• I question the focus on outcomes like mortality, as opposed to program outcomes which would include transmission. The model is already highly complex, but transmission seems like it would be an important consideration in this analysis. If mortality is relatively low, as it is for HIV negative patients, we won't see much of an effect. 

In the analysis presented here, we report the effect of the simulated interventions on a range of process (intermediate) and health outcomes, including True TB treated; Disability Adjusted Life Years Averted (DALYs) and Deaths (mortality). DALYs combine morbidity (time spent being unwell) as well as mortality. A summary of these results is provided in Table 3. 

The focus of this analysis is on the mechanisms leading to the translation of intermediate – to health outcomes and therefore the model was designed to show detail in the expression of those mechanisms. Additional model structure to include transmission, would have required compromise on some of the details of the processes around the pathways (including out of care) which we felt were important to retain as it speaks to the value of our model analysis which was based on empirical work. As authors of the study, we agree that a limitation of the study is that we did not include secondary benefits such as a reduction in tuberculosis transmission reduction. This limitation and implications influenced the time horizon of the model (restricted to 3 years) and are explained on page 8 (from line 5):

“Secondary benefits to the population due to tuberculosis transmission reduction are not included (29).”

And in more detail in the discussion section on page 24 (from line 12):

“The following limitations should be considered when interpreting our findings. Firstly, we did not model the effect of the various scenarios on the tuberculosis epidemic at a population level. While the implementation of Xpert primarily resulted in an increased identification of smear-negative tuberculosis, currently thought not to be a major driver of transmission, not including transmission in the analysis is likely to underestimate the relative benefit of reducing pre-treatment LTFU at a population level (65,75,76).”

Our argument is that if we included transmission dynamics in the model analysis, we are likely to have not had as much details on the identification of smear-negative tuberculosis among HIV-positive patients, which is thought not to be a major driver of transmission but is a major driver of mortality. 

• The authors state that improving diagnostic testing for patients with negative Xpert results would have an impact, reflecting higher mortality in HIV positive patients. However, it wasn't clear that results were quite sensitive to HIV rates in the population. 

The total DALYs per symptomatic individual entering the model in the baseline/ status quo arm of the study was 4.72. The results of the univariate sensitivity analyses are reported in Figure 4C in the main text and in the supplementary appendix (S1 Text) in Table 6 on page 33. If we increase the proportion of the cohort moving through the model who is HIV positive on ART from 0.155 to 1 meaning that the entire cohort is simulation is running as HIV positive individuals on ART, then the total DALYs increases to 5.36. This increase is offset slightly by a concomitant reduction in HIV positive patients not on ART who has higher mortality rates. It is important to remember that the prevalence of Tuberculosis in the empirical analysis as in the simulated cohort was 13%. Furthermore, we don’t see a higher increase in DALYs because the behaviour of healthcare workers when they see a patient who is HIV positive on ART and symptomatic do not scale in the model with increases as it might do in real life. Therefore, in the model, the largest shifts in the resulting outcomes come from changes in TB prevalence and changes in the behaviour of healthcare workers when they see a patient who is symptomatic and starts TB treatment after a test result. This interaction is visually shown in Figure 1 as the interactions at decision nodes A-C. The caption of Figure 4 has been updated to make the sensitivity analyses clearer and more accessible to the readers: 

“Figures 4(A), 4(B) and 4(C). Results from the univariate sensitivity analyses, showing the absolute value of outcomes at a low value of the parameter being changed (green) and at a high value of the same parameter (blue). The graph is a summary of the parameters with the greatest influence on the (A) provider cost, (B) the societal costs, and the (C) effectiveness (DALYs) of the base case (Xpert). “

• I also question complete lack of exploration of MDRTB rates and outcomes, and very low treatment rates, as Xpert should impact management of those cases. 

In the empirical work that informed the parameter estimation for the modelling study, the prevalence of rifampicin resistance was too low to assess the effect of Xpert MTB/RIF introduction. Similarly, to inform the simulation study, we did not have similar levels of data to inform parameterisation of this part of the model with a positive Xpert MTB/RIF test with Rifampicin resistance in 4.0% (8/200) of participants. We discuss this limitation on page 24, from line 19:

“Lastly, while our model includes a pathway for patients to initiate multi-drug resistant tuberculosis treatment if diagnosed, and incur the associated costs, we do not attempt to estimate the true prevalence of multi-drug resistant tuberculosis or what effect the investments may have on the epidemic. In the analysis of the negative pathway, therefore, the model may be underestimating the effect of incorrectly starting an individual on drug-sensitive tuberculosis. Studies following the roll-out of Xpert have found that barriers to initiating multi-drug resistant tuberculosis persisted and that the time-to-appropriate-treatment was only slightly reduced (8,77).”

• There are aspects of this analysis that would be very difficult to quantify, like resources that would support physician decision making. We don't have an idea of any downstream effects of the intervention, which limits the impact of this study.

There is certainly scope for further work in this area in the future but given that previous work in this area has been limited our work is a steppingstone for others to build on and to try further approaches. We acknowledge the limitations of being unable to model the downstream effects of the intervention on page 24, line 14. 

“Secondly, while we are modelling scenarios and benefits in a nuanced way, the relationship between health system structures, health care worker -, and patient behaviour is complex and while one can observe patterns, predictions will be limited by our understanding of the mechanisms driving these patterns. Thirdly, any investment in the health system will be likely to have an impact on other associated services (externalities), the benefits of which we did not include in our analysis (42).”

Furthermore, we amended the discussion of the challenges in estimating the costs and outcomes of more distal investments in the manuscript (page 23, line 27) as follows: 

“The outcomes of these investments are also likely to influence other disease programs and sectors (73). We do not include these spills over benefits or costs in our analysis, and thus our estimates are conservative. Empirical work has highlighted the importance of going beyond investing in assets and technology to invest in developing agency and governance (the software capacities of health systems) (74). Those investments are highly contextual and difficult to cost, so while our approach highlights to decision makers the resource envelopes required, more work is needed to develop and iteratively assess context-specific investment strategies. In-depth qualitative work to understand the barriers and facilitators of health care workers’ implementation of diagnostic guidelines would fill some of this gap.”

Notably an interesting externality of the decision to implement the Xpert platform in South Africa, borne out during the COVID pandemic in that the PCR platform, with COVID cartridges could be used to test samples for SARS COVID. One may argue that some externalities would be hard to anticipate though and that the research question (and policy makers’ questions) at the time defines the focus of the decision problem and outcomes to be included in the analysis.

 

Reviewer #4: 

The study is fascinating and concerns a typical example of the importance of implementation science, or translational science. In this case, it has been shown in several studies that Xpert is more sensitive and specific than the microscope in diagnosing tuberculosis. A more specific and sensitive tool allows identifying more cases and starting treatment earlier, with a health benefit and also an economic benefit. This is suggested, for example, in the study conducted by Orlando et al. (Orlando S, Triulzi I, Ciccacci F, et al. Delayed diagnosis and treatment of tuberculosis in HIV + patients in Mozambique: A cost-effectiveness analysis of screening protocols based on four symptom screening, smear microscopy, urine LAM test and Xpert MTB / RIF. 2018; Plos ONE 13 (7): e0200523). However, this evidence derived from trials or simulations is not confirmed in a pragmatic trial, that explore the use of the apparently best technology in the real world. The authors rightly hypothesize that the problem does not lie in the instrument (Xpert) but in its use by physicians and its ineffective inclusion within the diagnostic-therapeutic path. Therefore, it is exciting to evaluate how an additional investment in these aspects can modify the effectiveness and cost-effectiveness of the most advanced technology compared to the status-quo (the microscope). 

Thank you.

However, although the problem is very clear, in the course of the analysis, the fundamental steps concerning the issue in question are not clear, at least to the reader.

• It is said that the question is relational and studies in the field of sociology are mentioned, such as reference 23. In this case, it would have been necessary to approach the problem from a sociological point of view. Therefore, a qualitative analysis that analyzed in-depth the barriers to correct use of Xpert would be recommended. Instead, the issue is addressed from an economic point of view, assuming that the ineffective use of technology depends on some lack of investment.

Qualitative work to further understand especially health care workers’ barriers and facilitators to Xpert implementation would be valuable. We have added a statement to that effect to the discussion section of the manuscript (page 24, line 7), 

“In-depth qualitative work to understand the barriers and facilitators of health care workers’ implementation of diagnostic guidelines would fill some of this gap.”

• At this point, it would have been necessary to describe better what these investments are and what is their cost is. In the text, you never find this description nor an analysis of the costs of the 3 investments. Table 2 briefly describes the investments but does not explain in detail how these investments generate additional costs, what additional resources they need, and the cost of these resources.

Table 2 describes the implementation of the investment scenarios with a focus on the costs generated by changing probabilities in model parameters. The additional investment required to support the change is not known definitively due to a lack of empirical work however is included in a separate analysis presented in Figure 3. I have amended the paragraph on page 13, line 2 to flag the links between these analyses. 

“The pragmatic nature of the trial allowed us to identify gaps between ideal movement along different decision nodes of the pathway and mediating variables of effectiveness in routine care settings. Table 2 summarises the investment scenarios modelled and how they were implemented in the model. We modelled five investment strategies to support the tuberculosis diagnostic pathway. These included 1) reducing initial pre-treatment loss-to-follow-up (iLTFU), 2) supporting same-day clinical diagnosis of tuberculosis after a negative test result (TfN), and 3) improving access to further tuberculosis diagnostic tests following an initial negative result (NP). In addition, two combination scenarios were modelled (iLTFU and TFN; iLTFU and NP) to observe the additive effects of the scenarios. Investments were modelled by altering parameters at key stages in the patient pathway and how these will increase the count of utilisation that increases costs and affects outcomes. The cost of facilitating change through changing behaviour, which we refer to as the transaction cost is shown in Figure 3. “

• In practice, all cost analysis is deferred to other studies cited, especially Foster 2015 and Vassall 2017. But if this is the fundamental point of this analysis, and this aspect should be reported also in this study.

The cost analysis is explained in more detail in the Supplementary Appendix page 28, which includes Table 5 summarising the unit costs used and their sources. Including the following explanation:

“Costs incurred by the cohort simulated in the model is estimated by adding a unit cost value multiplied by the utilisation associated with a specific health state or process during the specified time interval. Costs were assessed from a societal perspective and are reported in 2013 US dollars (USD). Using similar arguments to ones posed by Meyer-Rath et al. 2015, costs were not inflated to present values (37), because adjusting for inflation would not accurately represent the present value of resource inputs given that some of the inputs do not track the consumer price inflation (CPI) index. For example, the prices of medicines used in the public sector are decided through a tendering process and then set for a number of years (38). Notably, human resource costs, the main driver of most of these unit costs are negotiated with labour unions, during this negotiation, the trajectory of increases are set. 

Unit costs were estimated as part of primary data collection alongside the trial, sampled from the same study sites where the outcomes data were collected. The details of the methodology associated with these costing studies have been published (27,39–41). A combination of top-down and bottom-up costing methods were used. So, for example, facility overhead costs were allocated to specific processes using a utilisation or staff time allocation factor. Processes were observed, inputs noted and valued, and interactions timed to estimate the unit cost of a procedure or input (40). 

Provider costs, the cost of diagnosing and treating patients with TB, were estimated for eight of the primary health care facilities included in the study (two per province) and included the cost of building health care facilities, the cost of human resources, the cost of any observed resources used and the cost of medication. The cost of medication was estimated from the South African Department of Health medicines price registry, which lists the tender price of medicines negotiated. We added 8% of the tender price of the medicine, to this cost for the distribution system (ref Margaret von Zeil, personal communication). For MDR TB treatment, we followed the estimates of Sinanovic and colleagues who constructed a cost of RR TB treatment by assuming a mixture of centralised and decentralised models of care were used nationally based on a 54%: 46% urban-rural split (41). Inpatient care for MDR treatment was assumed to be 44 days in the fully decentralised model and 128 days in the fully centralised model. The cost associated with Xpert in the laboratory was likewise calculated from primary data collection in twenty laboratories during test implementation, and includes the cost of laboratory space used to process the test, human resource costs (based on time spent processing observed), as well as the cost of any resources needed to conduct the required assays (40). Provider unit costs used are summarised in Table 5. 

The costs incurred by patients were estimated from patient exit interviews conducted with two cohorts of patients, in ten of the XTEND study clinics (27,39). The unit of analysis was the patient within their household and community. The first cohort of 351 people with suspected TB were interviewed at the time of receiving a TB diagnostic test and followed up six months later. The second cohort, 168 patients on TB treatment were recruited from the same ten facilities and followed up at five months on treatment. In addition, 134 RR TB patients at different stages in their treatment were interviewed with 82 of these receiving inpatient care and 52 receiving treatment in outpatient facilities. We estimated health care utilisation, out of pocket costs incurred due to transport and other expenses incurred. We also estimated patients’ income and income loss associated with ill health; as well as the cost of informal care. Cost results are presented separately for patient costs and ‘community costs’ that includes the cost of informal care.The costs associated with health seeking behaviour and time loss from the start of TB associated symptoms to getting tested for TB were estimated. The number of health service visits associated with receiving health care during case finding and treatment was based on a combination of patient reported (patient surveys), and provider reported visits for each facility. 

Where intervention scenarios modelled increased ART uptake, we included a monthly cost of ARV treatment and associated patient costs from secondary data sources. However, we do not include the cost of ART in all comparators. In the trial population, the implementation of Xpert did not increase the proportion of patients starting ART when compared against the smear microscopy arm of the study (8) and it is likely that adding the cost of ART could make interventions that differentially benefit those who are HIV negative appear more cost-effective (due to the significantly lower costs) than interventions that benefit patients on ART, with potential equity implications in the distribution of resources (42). The use of health services as patients progressed through care was collected as part of the trial through case note abstractions of identified fields in the patient records. In addition, patients were asked specific questions about their use of health services during their illness and care seeking.”

Given the word count limit, it was not feasible to include further detail in the main text of the manuscript.

• Also, from the point of view of effectiveness, it is not clear the mechanism through which these investments increase effectiveness, or rather allow to avert DALYs. Presumably through a reduction in mortality linked to an earlier identification of positive subjects, and a quicker start of treatment. However, these passages are not reported in the methodology and in the discussion, where I expected them to be the most important aspect to discuss, together to what was said above about costs. In essence, the mathematical model is a bit of a black box in which a higher cost generate a health benefit, but it is not clear, at least if we stick to the text, how these costs and benefits are generated. I suggest making these aspects clear in the text and not only in the tables. Also, cost analysis should be described more in detail, and possibly discuss how other studies have estimated the cost of these interventions (if any).

The mechanisms by which the interventions are implemented through changing parameter values are summarised in Table 2 (page 14). The first column describes the investment, the second column summarises how this change was implanted in the model, with the third column giving the specific changes made to the parameters including the initial and final parameter values. In the final column any assumptions (implicit or explicit) made through the model structure and values of the model parameters are summarised.

• Figures are not explained in the text. For example, figure 1, is not clear how it fits in the discourse and how to interpret it.

Figure 1 is a visual representation of the model structure and is referred to as a visual aid to the viewer when reading Table 2. I have added additional text to make this clearer on page 7, line 20 and when referring to Table 2 on page 13, line 2.

“A simplified visual representation of the model and the decision points is shown in Figure 1 and is referred to in Table 2 (27).”

“Table 2 summarises the investment scenarios modelled and how they were implemented in the model, with a visual representation of the model and decision points provided in Figure 1.”

Further detail on the model structure is provided in the supplementary appendix from page 3.

• Finally, the conversion rate in USD should consider Power Purchasing Parity for cost generated in South Africa, such as costs incurred by patients.

Power purchasing parity is potentially a useful approach to presenting the value of costs in an analysis such as ours when multiple countries are compared. However, given that the analysis is focused on a single country (South Africa), it is more appropriate to convert to a single comparative country often used in other similar analysis as allows for easier comparison to other studies. We show the conversion rate used on page 12, line 23.

• In the table 3, all alternatives are compared with the base case scenario (Microscopy), but I would have reached them in order of increased effectiveness. In this way, if I'm not wrong, some alternative will be excluded because they are dominated by others with lower cost and higher effectiveness. I could be wrong about this, but with this presentation of results is not possible for the reader to clearly identify dominated or extendedly dominated alternatives.

Currently the results in Table 3 are shown in the order of increasing effectiveness on DALYs. We do not explicitly show present the results in terms of which scenarios are dominated in the table because the investment scenarios are not mutually exclusive. Scenarios 1, 2 and 4 are stand-alone scenarios, however scenarios 3 and 5 are combinatorial and presents the complexity of these investments. Additionally, the costs shown here are not the full costs from the perspective of the policy maker ad should be read with the transaction cost analysis in Figure 3.

---

## [Decision Letter · Decision Letter 2]

29 Apr 2021

Strengthening health systems to improve the value of Tuberculosis diagnostics in South Africa: a cost and cost-effectiveness analysis.

PONE-D-20-18039R2

Dear Dr. Foster,

We’re pleased to inform you that your manuscript has been judged scientifically suitable for publication and will be formally accepted for publication once it meets all outstanding technical requirements.

Kind regards,

Frederick Quinn

Academic Editor

PLOS ONE

Additional Editor Comments (optional):

Reviewers' comments:

Reviewer's Responses to Questions

**Comments to the Author**

1. If the authors have adequately addressed your comments raised in a previous round of review and you feel that this manuscript is now acceptable for publication, you may indicate that here to bypass the “Comments to the Author” section, enter your conflict of interest statement in the “Confidential to Editor” section, and submit your "Accept" recommendation.

Reviewer #4: All comments have been addressed

2. Is the manuscript technically sound, and do the data support the conclusions?

Reviewer #4: Yes

3. Has the statistical analysis been performed appropriately and rigorously? 

Reviewer #4: I Don't Know

4. Have the authors made all data underlying the findings in their manuscript fully available?

Reviewer #4: Yes

5. Is the manuscript presented in an intelligible fashion and written in standard English?

Reviewer #4: Yes

6. Review Comments to the Author

Reviewer #4: (No Response)

7. PLOS authors have the option to publish the peer review history of their article (what does this mean?). If published, this will include your full peer review and any attached files.

Reviewer #4: No

---

## [Editor Report · Acceptance letter]

3 May 2021

PONE-D-20-18039R2 

Strengthening health systems to improve the value of tuberculosis diagnostics in South Africa: a cost and cost-effectiveness analysis 

Dear Dr. Foster:

I'm pleased to inform you that your manuscript has been deemed suitable for publication in PLOS ONE. Congratulations! Your manuscript is now with our production department. 

Kind regards, 

on behalf of

Dr. Frederick Quinn 

Academic Editor

PLOS ONE